# Clinicopathological Features of Invasive Breast Cancer: A Five-Year Retrospective Study in Southern and South-Western Ethiopia

**DOI:** 10.3390/medicines10050030

**Published:** 2023-05-04

**Authors:** Esmael Besufikad Belachew, Adey Feleke Desta, Dinksira Bekele Deneke, Bizunesh Dires Fenta, Alemwosen Teklehaymanot Alem, Abdo Kedir Abafogi, Fekade Yerakly Lukas, Mesele Bezabih, Dareskedar Tsehay Sewasew, Eva J. Kantelhardt, Tesfaye Sisay Tessema, Rawleigh Howe

**Affiliations:** 1Biology Department, College of Natural and Computational Sciences, Mizan Tepi University, Addis Ababa 260, Ethiopia; 2Department of Microbial, Cellular and Molecular Biology, College of Natural and Computational Sciences, Addis Ababa University, Addis Ababa 1176, Ethiopia; 3Armauer Hansen Research Institute, Addis Ababa 1005, Ethiopia; 4Pathology Department, Hawassa Referral Hospital, Hawassa 1560, Ethiopia; 5Pathology Department, Jimma University Specialized Hospital, Jimma 378, Ethiopia; 6Institute for Medical Epidemiology, Biometry and Computer Science, Martin Luther University Halle-Wittenberg, 06097 Halle, Germany; 7Institute of Biotechnology, Addis Ababa University, Addis Ababa 1176, Ethiopia

**Keywords:** breast cancer, histological type, invasive ductal carcinoma, lymph node involvement, stage and grade

## Abstract

**Background:** Breast cancer (BC) is the most common type of cancer in Ethiopia. The incidence of BC is also rising, but the exact figure is still poorly known. Therefore, this study was conducted to address the gap in epidemiological data on BC in southern and southwestern Ethiopia. **Materials and Methods:** This is a five-year (2015–2019) retrospective study. The demographic and clinicopathological data were collected from biopsy reports of different kinds of breast carcinomas in the pathology department of Jimma University Specialized Hospital and Hawassa University Specialized Referral Hospital. Histopathological grades and stages were conducted using Nottingham grading and TNM staging system, respectively. Collected data were entered and analyzed using SPSS Version-20 software. **Results:** The mean age of patients at diagnosis was 42.27 (SD = 13.57) years. The pathological stage of most BC patients was stage III, and most of them had tumor sizes greater than 5 cm. Most patients had moderately differentiated tumor grade, and mastectomy was the most common type of surgery at the time of diagnosis. Invasive ductal carcinoma was the most common histological type of BC, followed by invasive lobular carcinoma. Lymph node involvement was seen in 60.5% of cases. Lymph node involvement was associated with tumor size (χ^2^ = 8.55, *p* = 0.033) and type of surgery (χ^2^ = 39.69, *p* < 0.001). Conclusions: This study showed that BC patients in southern and southwestern Ethiopia displayed advanced pathological stages, relatively young age at diagnosis, and predominant invasive ductal carcinoma histological patterns.

## 1. Background

Cancer is characterized by an uncontrolled and invasive growth of cells that spreads to other parts of the body through blood circulation and lymphatic vessels [1]. The sustaining proliferative signals, evading growth suppressors, resisting cell death, enabling replicative immortality, inducing angiogenesis, and activating invasion and metastasis are the main features of cancer [2].

There were an estimated 19.3 million new cases and 9.9 million deaths from cancer in 2020. A high percentage of breast cancer (BC) cases (11.7%), followed by lung (11.4%), colorectal (10.0 %), prostate (7.3%), and stomach (5.6%) cancers were reported. Lung cancer remained the leading cause of cancer death, with an estimated 1.8 million deaths (18%), followed by colorectal (9.4%), liver (8.3%), stomach (7.7%), and female breast (6.9%) cancers [3]. Breast cancer is one of the leading causes of mortality among women worldwide, with an estimated 2.2 million cases and 684, 996 deaths in 2020, accounting for 24.5% of cancer cases and 15.5% of cancer deaths [4,5]. It is predicted to cause more than 3 million new cases and 1 million deaths by 2040 [3,6]. The number of new cases in 2020 accounted for female breast cancer were as follows: China (18.4%), which had the highest number of new cases; United States of America (11.2%); India (7.9%); Japan (4.1%); Brazil (3.9%); the Russian Federation (3.3); Germany (3.1%); and all other countries (48.1%) [3]. The incidence of female breast cancer worldwide is increasing, with the growth rate higher in younger women in comparison to older women [7]. Breast cancer mortality is high in Africa, with the highest mortality rates being recorded in sub-Saharan Africa [8,9]. In 2020, there were 85,700 deaths in Africa, and there will likely be 163,000 deaths in 2040 [3]. The high number of deaths in Africa may be due to inadequate access to early detection, diagnosis, and treatment. However, it is also related to cultural differences in lifestyle behaviors, socioeconomic factors, and differences in the biological characteristics of breast cancer [10]. Breast cancer has also been shown to occur at an earlier age in African countries than in others, with a median age of approximately 45 years [4]. It also has a significant economic impact [11,12].

In Ethiopia, the number of cancer cases and deaths were estimated to be 77,352 and 51,865, respectively, in 2020. Breast cancer is the most common type of cancer in Ethiopia and constitutes 31.9% of total cancer cases in women, accounting for 16,133 new cases and 9061 deaths in 2020 [3]. In Ethiopia, BC mainly affects young women and shows an advanced stage at diagnosis [13]. The incidence of BC is also rising in Ethiopia [14], but the exact figure is still not known. The lack of appropriate data in southern and southwestern Ethiopia has hampered an accurate national estimate of the BC burden; hence, the objective of this study was to address the gap in epidemiological data on BC in southern and southwestern Ethiopia.

## 2. Materials and Methods

This is a retrospective study (2015–2019) based on data collected using reports from the pathology departments of Jimma University Specialized Hospital and Hawassa University Specialized Referral Hospital. Only histopathological diagnoses of BC from 2015 to 2019 were included in this study.

### 2.1. Data Collection

The demographic and clinicopathological data were collected from biopsy reports of the pathology department in each hospital using a data collection form. Demographic information, such as age, sex, and place, and clinicopathological data, such as histologic type, stage, tumor size, grade, lymph node involvement, number of involved lymph nodes, and nature of specimen, were included. Only 50 histopathology slides were inspected and reviewed at random.

### 2.2. Inclusion Criteria

Only invasive BC was included in this study. A total of 294 patients with invasive BC were included, and 34 patients with carcinoma in situ were excluded.

### 2.3. Exclusion Criteria

Breast cancer patients with pathologies of carcinoma in situ were excluded. Apart from this, 32 patients’ stage, 39 tumor sizes, 161 grades, 8 lymph node involvements, 11 numbers of involved lymph nodes, 4 types of surgery, and 6 histological-type patient data were omitted because the information was not available in their pathology reports. In addition to being absent on their medical records, tumor stage, tumor size, and lymph node involvement were not performed for every incisional biopsy. Moreover, in some cases of invasive BC, lymph node dissection was not always performed.

### 2.4. Classification Criteria

Histopathological grades were measured using Nottingham grading. The evaluation of tubule formation, nuclear pleomorphism, and mitotic counts, each of which is scored 1–3, were used to determine the grade for each particular breast tumor. The scores for each category were added together to calculate the overall tumor grade, which has a range of 3–9. The following criteria were used to assess tumors: well-differentiated or grade I scores of 3–5; moderately differentiating or grade II scores of 6–7; and poorly differentiating or grade III scores of 8–9 [15]. The TNM staging method was used to perform histopathological stages. This method provides a “stage grouping” based on the extent of the primary tumor (T), local lymph nodes (N), and distant metastases (M) [16] (Table 1).

### 2.5. Data Analysis

Data collected from pathology reports were entered and analyzed using SPSS Version-20 software. Statistical significance was defined as a *p*-value of less than 0.05, and all statistical tests were two-sided. For categorical measures, frequencies and percentages were examined. For continuous measures, mean, standard deviation, and range were examined. Fisher’s exact test was used to determine association.

## 3. Results

### 3.1. Demographic Characteristics

A total of 589 patients presented with breast lumps at Hawassa University Comprehensive Specialized Referral Hospital and Jimma University Specialized Teaching Hospital from 2015 to 2019. Of these, 294 patients (49.9%) had a histopathologically confirmed diagnosis of invasive breast carcinoma. The 50 histopathology slides that were inspected and reviewed agreed with the recorded data that had been collected. The age at diagnosis of BC patients ranged from 17 to 100 years with a mean of 42.27 (SD = 13.57 years). The majority of BC patients (61.6%) were under 44 years of age (Table 2).

### 3.2. Clinicopathological Features

The pathological stage of the majority of BC patients (64.9%) was stage III, and 41.2% of the patients had a tumor size greater than 5 cm (pT3) at the time of diagnosis. It was observed that most patients (49.6%) had a moderately differentiated tumor grade (grade II) followed by poorly differentiated (grade III) (28.6) and well-differentiated (grade I) (21.8%) grades. Lymph node involvement was seen in 60.5% of cases, and most of them (48.8) had 4–10 involved lymph nodes. Mastectomy was the most common method of surgery, accounting for 84.5% of cases (Table 3).

Invasive ductal carcinoma was the most common histologic type of BC (77.1%) followed by invasive lobular carcinoma (11.1%) and papillary carcinoma (2.8%) (Figure 1). The representative picture of each histological type is shown in Figure 2.

Lymph node involvement was significantly associated with tumor size (χ^2^ = 8.55, *p* = 0.033) and type of surgery (χ^2^ = 49.09, *p* < 0.001), with higher lymph node involvement observed in patients undergoing mastectomy. Patients with lymph node involvement had a higher proportion of tumors with a size greater than 5 cm (46.6%) compared with those without lymph node involvement. Lymph node involvement was also higher among invasive ductal carcinoma histologic types (Table 4).

## 4. Discussion

In this study, the mean age at diagnosis of BC was 42.27 years. The mean age in this study was comparable with previous studies conducted in Ethiopia by Gemta et al. (44 years), Kantelhardt et al. (43 years), and Ersumo (44.77 years) [13,17,18]. Similar findings were also reported from other African counties with a mean age ranging from 45.8 to 48.7 years [19,20,21,22,23]. In contrast to these studies, the mean age of BC patients at diagnosis is higher in high-income countries [24].

Most patients (64.9%) were diagnosed with TNM stage III BC in this study. This finding is similar to other studies conducted in Ethiopia, Pakistan, and Haiti, reporting 56.7%, 57.4%, and 55.5% stage III BC, respectively [13,25,26]. In contrast, early-stage presentation is more commonly reported in Europe and India [24,27]. The study conducted in Ethiopia also revealed that only a small proportion of the women were found to perform breast self-examination and clinical examination [28,29], and this could be the possible reason for having advanced-stage BC in this study. Lack of awareness and screening could also be plausible reasons.

Moderately differentiated (grade II) BC was found in most (49.6%) patients. This result agreed with other studies conducted by Gemta et al., Hadi and Jamal, Takalkar et al., Pathak et al., Toma et al., Malanda et al., Adeniji et al., Ezike et al., Zilenaite et al., Uyisenga et al. and Maffuz-Aziz et al., who reported 46.2%, 47.3%, 75.4%, 76.8%, 53.8%, 51,3%, 52.6%, 59.5%, 46.5%, 52.9%, and 54.1% of grade II tumor, respectively [13,25,27,30,31,32,33,34,35,36,37]. Conversely, poorly differentiated (grade III) BC was common in another study conducted in Ghana (49%) [30]. Grade III is the most common grade in both African and African-American populations, but European descent has lower grades, which may be related to racial disparities, lower socioeconomic status, environmental or lifestyle factors, and late diagnosis in the African population [38].

Lymph node involvement at the time of diagnosis was seen in most BC patients (60.5%) in this study, which is in agreement with other studies conducted by Gemta et al., Rambau et al., Adani-Ifè et al., Adeniji et al. and Hadi and Jamal which had 62.4%, 75.5%, 75.2%, 89.2%, and 90% lymph node involvement, respectively [13,20,25,34,39]. In contrast to this study, 41.2% of lymph node involvement was seen in Mexico [31], probably due to the earlier-stage diagnoses in this country compared to Ethiopia.

In this study, lymph node involvement showed the highest frequency among invasive ductal carcinoma (78.9%). Another study conducted by Balekouzou et al. (87.5%) reported similar findings [19]. Involvement of lymph nodes was also significantly associated with tumor size (χ^2^ = 8.55, *p* = 0.033), and the highest frequency of lymph node involvement was seen in tumors with sizes greater than 5 cm (46.6%). This result was in line with the study conducted by Hadi and Jamal in Pakistan, in which 75.4% of lymph nodes were involved [25]. The exact estimation of the size of the tumor is necessary before surgery to make the best decision for the management of patients [40]. Lymph node involvement was also significantly associated with the type of surgery (χ^2^ = 39.69, *p* < 0.001), with the highest frequency among mastectomy types of surgery (95.9%).

Globally, more than 75% of breast carcinoma is histologically invasive ductal carcinoma [41]. Similarly, invasive ductal carcinoma was the most common (77.1%) histologic type of BC in this study. This result is compatible with studies conducted by Hailu et al. (81.6%) in Ethiopia, Kantelhardt et al. (79.2%) in Ethiopia, Ersumo (77.6%) in Ethiopia, Pathak et al. (75%) in Nepal, Maffuz-Aziz et al. (79.7%) in Mexico, and Oluogun et al. (88.9%) in Ghana [14,17,18,31,42,43].

In this study, tumor sizes of more than 5 cm (pT3) were most common (41.2%) at the time of diagnosis. In a study conducted in Ghana [44], the five-year mortality rate among BC patients with a tumor size of more than 5 cm was 48% more than patients with tumor sizes of less than 5 cm, indicating the predictive importance of tumor size.

Mastectomy was the most commonly performed surgery, accounting for 84.5% of BC patients in this study. In contrast, lumpectomy has been the most common (69%) in Europe. This could be due to the early detection/presentation of patients with smaller tumor sizes, the practice of breast-conserving surgeries, and better diagnostic techniques. The mastectomy rate in Africa is more than 85%, compared to 30% in Europe [24].

## 5. Conclusions

This study shows that BC in southern and southwestern Ethiopia is observed in a relatively young age group, with a predominant histologic pattern of invasive ductal carcinoma. A high rate of lymph node involvement, advanced pathological stage, and grade at the time of diagnosis was reported. Lymph node involvement has a significant association with tumor size and type of surgery. A lack of awareness of BC self-examination and screening, poor health care system, shortage of expertise, and the aggressive nature of tumors may contribute to the occurrence of advanced disease at the time of diagnosis.

## Figures and Tables

**Figure 1 medicines-10-00030-f001:**
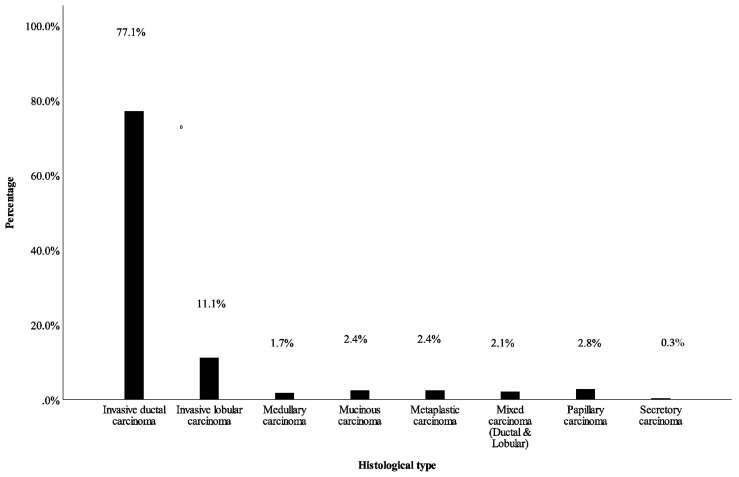
Distribution of histologic types of BC. Data pooled from Hawassa University Comprehensive Specialized Referral Hospital and Jimma University Specialized Teaching Hospital.

**Figure 2 medicines-10-00030-f002:**
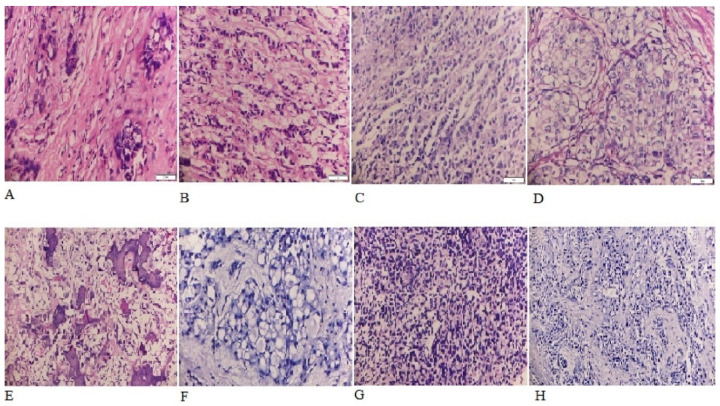
Histological types of breast cancer: (**A**) Invasive ductal carcinoma; (**B**) Invasive lobular carcinoma; (**C**) Mixed (invasive ductal and lobular) carcinoma; (**D**) Medullary carcinoma; (**E**) Metaplastic carcinoma; (**F**) Mucinous carcinoma; (**G**) Papillary carcinoma; (**H**) Secretory carcinoma.

**Table 1 medicines-10-00030-t001:** The staging methods for breast cancer tumors.

Stages	Tumor Size (T)	Regional Lymph Nodes (N)	Distant Metastases (M)
0	Tis	N0	M0
IA	T1 *	N0	M0
IB	T0	N1mi	M0
	T1 *	N1mi	M0
IIA	T0	N1 **	M0
	T *	N1 **	M0
	T2	N0	M0
IIB	T2	N1	M0
	T3	N0	M0
IIIA	T0	N2	M0
	T1 *	N2	M0
	T2	N2	M0
	T3	N1	M0
	T3	N2	M0
IIIB	T4	N0	M0
	T4	N1	M0
	T4	N2	M0
IIIC	Any T	N3	M0
IV	Any T	Any N	M1

* T1 includes T1mi. ** Only T0 and T1 tumors with nodal micrometastases are excluded from Stage IIA and are instead classified as being Stage IB (Source: [16]).

**Table 2 medicines-10-00030-t002:** Demographic characteristics of BC patients at Hawassa University Comprehensive Specialized Referral Hospital and Jimma University Specialized Teaching Hospital from 2015 to 2019.

Variables			*n* (%)
Age group		15–29	45 (15.3%)
		30–44	136 (46.3)
		45–59	77 (26.2)
		60–74	28 (9.5)
		≥75	8 (2.7)
		Total	294 (100.0)
Mean ± Sd (Minimum, Maximum) 42.27 ± 13.57 (17,100)			
Sex	Female		284 (96.6)
	Male		10 (3.4)
	Total		294 (100.0)
Patient residence	Hawassa		113 (38.4)
	Jimma		181 (61.6)
	Total		294 (100.0)

**Table 3 medicines-10-00030-t003:** Distribution of tumor presentation at the time of diagnosis among BC patients at Hawassa University Comprehensive Specialized Referral Hospital and Jimma University Specialized Teaching Hospital from 2015 to 2019.

Variables		*n* (%)
Stage	I	5 (1.9)
	II	87 (33.2)
	III	170 (64.9)
	Total	262 (100.0)
Tumor Size	pT1	8 (3.1)
	pT2	71 (27.8)
	pT3	105 (41.2)
	pT4	71 (27.8)
	Total	255 (100.0)
Grade	I	29 (21.8)
	II	66 (49.6)
	III	38 (28.6)
	Total	133 (100.0)
Lymph node involvement	Positive	173 (60.5)
	Negative	113 (39.5)
	Total	286 (100.0)
Positive lymph nodes at surgery	1–3	70 (43.2)
	4–10	79 (48.8)
	>10	13 (8.0)
	Total	162 (100.0)
Type of surgery	Mastectomy	245 (84.5)
	Lumpectomy	2 (0.7)
	Incisional biopsy	43 (14.8)
	Total	290 (100.0)

**Table 4 medicines-10-00030-t004:** Lymph node involvement with stage, tumor size, and histologic type.

Variables	*n* (%)	Lymph Node Involvement		χ^2^	*p*-Value
		Yes	No		
Tumor Size, *n* (%)					
≤2 cm	8 (3.1)	3 (1.9)	5 (5.4)	8.55	0.033
>2 cm and ≤5 cm	70 (27.6)	37 (23.0)	33 (35.5)		
>5 cm	105 (41.3)	75 (46.6)	25 (26.9)		
Any size extension to the chest wall or skin	71 (28.0)	46 (28.6)	25 (26.9)		
Total	254 (100.0)	161 (100.0)	93 (100.0)		
Type of surgery, *n* (%)					
Mastectomy	242 (85.5)	165 (95.9)	77 (69.4)	39.69	< 0.001
Lumpectomy	2 (0.7)	1 (0.6)	1 (0.9)		
Incisional biopsy	39 (13.8)	6 (3.5)	33 (29.7)		
Total	283 (100.0)	172 (100.0)	111 (100.0)		
Histologic type, *n* (%)					
Invasive ductal carcinoma	218 (77.3)	135 (78.9)	83 (74.8)		
Invasive lobular carcinoma	32 (11.3)	19 (11.1)	13 (11.7)	7.4	0.37
Mixed carcinoma	6 (2.1)	4 (2.3)	2 (1.8)		
Papillary carcinoma	7 (2.5)	2 (1.2)	5 (4.5)		
Mucinous carcinoma	6 (2.1)	2 (1.2)	4 (3.6)		
Metaplastic carcinoma	7 (2.5)	5 (2.9)	2 (1.8)		
Medullary carcinoma	5 (1.8)	4 (2.3)	1 (0.9)		
Secretory carcinoma	1 (0.4)	0 (0.0)	1 (0.9)		
Total	282 (100.0)	171 (100.0)	111 (100.0)		

## Data Availability

The raw data in the SPSS file under the identification policy can be provided, for research purposes only, upon request via e-mail to the corresponding author.

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
