# Peer review of "Clinicopathological Features of Invasive Breast Cancer: A Five-Year Retrospective Study in Southern and South-Western Ethiopia"

_medicines, 2023, doi:10.3390/medicines10050030_

Round 1

Reviewer 1 Report

The paper gives an overview on a retrospective study regarding the statistic of the different kinds of breast carcinomas in Ethiopia. Although the study has substantial quantity of data I think some aspects can be improved. 

Please refer to the comments:

Overview: the paper gives an overview on a retrospective study regarding the statistic of the different kinds of breast carcinomas in Ethiopia. Although the study has substantial quantity of data I think some aspects can be improved.

In the Data Collection paragraph, you mentioned the inspection of 50 histopathological slides, but they are not reported in the paper. Addin the slides picture and analysis would be helpful and give more in-depth to the study.

You always refer to invasive breast carcinomas, of what molecular subtype are you talking about? The most aggressive and invasive breast carcinoma is the so called triple negative breast cancer. Are all the cases triple negative? I think you should specify which kind you’re taking in consideration.

General comments: the histogram needs to be uploaded at higher resolution. Overall the article is well written and covers an interesting subject. 

Author Response

Please see the attachment. Your comment is really crucial for the improvement of this manuscript.  The updated manuscript is also uploaded to the system.

Reviewer 2 Report

The study presents data on breast cancer in Southern and South-western Ethiopia.

1)      Authors mention the random inspection of 50 histopathology slides but this does not seem to be mentioned in the results section, did this random section reveal agreement or disagreement with the recorded data?

2)      The title Histomorphologic features of invasive breast cancer: a five years retrospective study in Southern and South-Western Ethiopia” is very suggestive of a strict focus on Histomorphologic features, while the actual data also provide a number of clinicopathological features such as surgery type, tumour size, lymph node involvement   etc.. The authors may thus consider to rephrase the title.

3) Please introduce abbreviations, even in abstract, for instance, breast cancer (BC) in the first sentence.

Writing is satisfactory but grammar correction at least by a digital service such as Grammarly is recommended, for instance:

4)      “In Ethiopia, the estimated number of cancer cases and deaths is 67,573 and 47,954, respectively in 2018.” - The past results should be described in past tense, therefore WAS instead of IS and a comma needs to be introduced after RESPECTIVELY. Please correct throughout the manuscript text

5)       Similarly, in: “2.2. Inclusion criteria Invasive BC is only included.” IS should be replaced by WAS.

6)       The large section of the results below does not refer to either a figure or a table, I assume that the text there should refer to Table 3 in each sentence:

Lymph node involvement was significantly associated with tumor size (χ2 = 8.55, p = 0.033), and type of surgery (χ2 = 49.09, p < 0.001). Patients with lymph node involvement had a higher proportion of tumors with a size greater than 5 cm (46.6%), compared with those without lymph node involvement. Lymph node involvement was also higher among invasive ductal carcinoma histomorphological types.

7) It would be good to update the     reference list with more studies done in 2022 and 2023

Author Response

(The authors gave the same response as above.)

Round 2

Reviewer 1 Report

All the comments and recommendations from previous review have been satisfied.